# A Tunable Terahertz Metamaterial Absorber Composed of Hourglass-Shaped Graphene Arrays

**DOI:** 10.3390/nano10030533

**Published:** 2020-03-17

**Authors:** Yunping Qi, Yu Zhang, Chuqin Liu, Ting Zhang, Baohe Zhang, Liyuan Wang, Xiangyu Deng, Xiangxian Wang, Yang Yu

**Affiliations:** 1College of Physics and Electronic Engineering, Northwest Normal University, Lanzhou 730070, China; 18215151763@163.com (C.L.); tingzhang718@126.com (T.Z.); baohezhangemail@126.com (B.Z.); wangliyuan122@126.com (L.W.); dengxy000@126.com (X.D.); 2Engineering Research Center of Gansu Provence for Intelligent Information Technology and Application, Northwest Normal University, Lanzhou 730070, China; 3School of Science, Lanzhou University of Technology, Lanzhou 730050, China; wangxx869@126.com; 4School of Civil and Environmental Engineering, University of Technology Sydney, Sydney, NSW 2007, Australia; yang.yu@uts.edu.au

**Keywords:** graphene, metamaterial absorber, surface plasmon resonance, finite difference time domain

## Abstract

In this paper, we demonstrate a tunable periodic hourglass-shaped graphene arrays absorber in the infrared (IR) and terahertz (THz) frequency bands. The effects of graphene geometric parameters, chemical potentials, periods, and incident angles on the pure absorption characteristics are studied by using the Finite Difference Time Domain (FDTD) method. In addition, this paper also analyzes the pure absorption characteristics of bilayer graphene arrays. The simulation results show that the maximum absorption reaches 38.2% for the monolayer graphene structure. Furthermore, comparing the bilayer graphene structure with the monolayer structure under the same conditions shows that the bilayer structure has a tunable dual-band selective absorption effect and has a higher maximum absorption of 41.7%. Moreover, it was found that there are dual-band tunable absorption peaks at 21.6 μm and 36.3 μm with the maximum absorption of 41.7% and 11%. The proposed structure is a convenient method which could be used in the design of graphene-based optoelectronic devices, biosensors, and environmental monitors.

## 1. Introduction

Graphene is a two-dimensional (2D) carbon nanomaterial with a hexagonal honeycomb structure composed of carbon atoms and sp^2^ hybrid orbitals [1,2]. Owing to its unique optoelectronic properties, graphene is one of the most promising optoelectronic materials and is widely used in the field of batteries, materials processing, biomolecular sensing, food safety, and communications [3,4,5,6,7,8,9,10,11,12,13,14,15,16,17]. Additionally, graphene also play an essential role in metadevices, such as detectors and modulators [18,19,20]. Taking the modulator as an example, the application of graphene enables the modulation depth of the modulator to reach 100%.

In the infrared (IR) and terahertz (THz) range, when graphene interacts with incident light [21,22,23,24,25,26,27,28], surface plasmon polaritons (SPPs) and localized surface plasmons (LSPs) occur. SPPs are surface waves that are excited at the boundary of the material; the excitation of these charge waves is achieved by properly matching the free space and surface plasmon momenta of the system. On the other hand, LSPs are subwavelength surface waves supported in materials whose characteristic dimensions are comparable to the excitation wavelength [29,30]. It is the latter that contributes to the absorption mechanism, and leads to the enhancement of absorption [31,32,33,34,35]. The migration of graphene electrons is described by the interband and intraband contributions, which are affected by the external electric fields and magnetic fields. The electromagnetic properties of graphene can be easily tuned by applying external electric fields, magnetic fields, or by chemical doping [36,37]. All of these can make the properties of graphene better than metal [38]. The use of graphene localized surface plasmon characteristics can effectively control the absorption and transmission of light, making graphene attractive to researchers.

In the past few years, graphene-based absorbers (GBA) have received much attention. In 2015, Ke et al. reported a cross-shaped graphene array absorber that achieved 20% absorption [39]; Xiao et al. proposed periodic graphene ring arrays and introduced a good angular polarization tolerance that achieved an absorption of 25% [40]; Fang et al. reached 30% absorption by incorporating graphene nanodisk arrays into an active device [41]. Although the maximum absorption of monolayer graphene has been greatly improved compared with its predecessors, the maximum absorption of a pure graphene layer is not more than 30%. Therefore, designing a graphene absorber with a higher absorption is an urgent problem to be solved. We compared our work with [42], which achieves ultrabroadband and nearly 100% perfect absorption by etching a cross-shaped structure on doped silicon. However, the substrate and patterned metamaterial of our proposed structure are different from those of [42], and for this reason, the absorption rate of our proposed structure rarely exceeds 50% [43]. When compared with [40,41], our structure has the following advantages: the structure has more geometric structures that can be optimized, so that we have more options to adjust its absorption characteristics; the structure is easy to integrate and the substrate of this structure is silicon and silica.

On the basis of these points, this paper proposed a GBA which is composed of periodically patterned “hourglass” graphene metamaterial arrays, and the proposed structure can reach an absorption of 41.7%. For a symmetric dielectric environment, the predicted maximum absorption does not exceed 50% [43], so the maximum absorption of 41.7% in this structure is still a remarkably good result. If a layer of golden mirror is added to the bottom of the structure, our proposed structure can also realize perfect absorption (Perfect Absorption in the Appendix A, shown in Figure A1).

The hourglass-shaped structure consists of two isosceles triangles and two semi-ellipses. These two triangles are symmetrical and intersect with each other. The reason why this structure was chosen are as follows: (1) the two symmetrical and intersecting triangles can realize zero-energy states; (2) the semi-ellipses have two geometric parameters that can be tuned, we can adjust more geometric parameters to adjust its absorption characteristics; (3) the two triangles and two semi-ellipses consist of a continuous graphene structure. In the second section, the hourglass-shaped structure is presented, we also introduce the calculation methods of the absorption spectra. In the third section, we study the influence of different parameters on the absorption characteristics. Then, we analyze the effects of bilayer graphene on absorption. Finally, we summarize the whole project.

## 2. Geometric Structures and Methods

The schematic of the tunable hourglass-shaped GBA is shown in Figure 1a. The period of the cell structure along the *X* and *Y* directions is *P*. There are two semi-ellipses and the semimajor and semiminor axis of the semi-ellipse are *L* and *R*. The middle part of the graphene structure is composed of two symmetrical isosceles triangles that intersect at the vertices. The graphene arrays are attached to a silicon (Si) substrate separated by a thin silica (SiO_2_) spacer layer with a thickness of $d_{1}$. In this paper, the absorption spectra and the localized electric field distribution of the structure are studied using the Finite Difference Time Domain (FDTD) method [44]. The antisymmetric, symmetric, and perfectly matched layer (PML) boundary conditions are used in the *X*, *Y,* and *Z* directions, respectively. The isosceles triangle of this structure has two sides of length 0.8 μm. The semimajor (*L*) and semiminor (*R*) axis of the ellipse are both 0.4 μm (that is the special case of an ellipse: circle). The thickness of SiO_2_ ($d_{1}$) and Si are 0.3 μm and infinity. The period (***P***) of the structure is 3 μm. The thickness of monolayer graphene is 0.34 nm. The uniform mesh accuracy along the *X* and *Y* directions are adopted as 20 nm, and the *Z* direction is 1 nm. The relative permittivity of Si and SiO_2_ are adopted as 1.96 and 3.9 [45,46]. The entire system is illuminated by a plane wave propagating along the negative *Z* direction with total electric field ***E*** polarizing along the *X* direction. The position of the electric field monitor is placed at the same position of the graphene layer and is larger than the graphene layer boundary.

The chemical potential ($\mu_{c}$) can be dynamically tuned by changing the value of total carrier density (N) that can be expressed as follows [47,48,49,50,51]:(1)$$\left|\mu_{c}\right|=\hbar v_{F}{\left(\pi N\right)}^{1/2}$$
where ħ and $v_{F}$ are the reduced Planck’s constant and the Fermi velocity [52]. The total carrier density (N) can be changed by applying a voltage ($V_{b}$), so N can be expressed as
(2)$$N=\varepsilon_{0}\varepsilon_{r}\left|V_{b}\right|/ed_{1}$$
where $\varepsilon_{0}$ and $\varepsilon_{r}$ are the permittivity of free space and the dielectric layer, $d_{1}$ and *e* are the thickness of the SiO_2_ layer and charge of an electron, respectively. Therefore Equation (1) can be expressed as follows:(3)$$\left|\mu_{c}\right|\approx \hbar v_{F}{\left(\pi \varepsilon_{0}\varepsilon_{r}\left|V_{b}\right|/ed_{1}\right)}^{1/2}\propto V_{b}{}^{1/2}$$

It can be seen from Equation (3) that with other parameters are fixed, and $\mu_{c}$ is related to $V_{b}$. When a $V_{b}$ is employed between the top and back gates (as shown in Figure 1b; the ion-gel layer is the conductive layer), N and $\mu_{c}$ can be dynamically tuned. We can change N by applying a specific $V_{b}$, then the change of N leads to the change of $\mu_{c}$, and we can obtain a specific value of $\mu_{c}$.

The conductivity of monolayer graphene can be calculated with the random phase approximation (RPA) in the local limit, consisting of intraband and interband contributions [53,54,55], which can be expressed as follows:(4)$$\sigma_{g}(\omega )=\sigma_{\mathrm{int}\mathrm{ra}}(\omega )+\sigma_{\mathrm{int}\mathrm{er}}(\omega )$$
(5)$$\sigma_{\mathrm{int}ra}(\omega )=\frac{i}{\omega +i\tau^{-1}}\frac{2e^{2}k_{B}T}{\pi \hbar^{2}}\ln\left[2\cos(\frac{\mu_{c}}{2k_{B}T})\right]$$
(6)$$\sigma_{\mathrm{int}er}(\omega )=\frac{e^{2}}{4\hbar }\left[\frac{1}{2}+\frac{1}{\pi }\arctan\left(\frac{\hbar \omega -2\mu_{c}}{2k_{B}T}\right)-\frac{i}{2\pi }\ln\frac{{\left(\hbar \omega -2\mu_{c}\right)}^{2}}{{\left(\hbar \omega -2\mu_{c}\right)}^{2}+4{\left(k_{B}T\right)}^{2}}\right]$$
where $k_{B}$ is the Boltzmann constant, ω is the angular frequency of the electromagnetic wave, *T* is temperature and is fixed at 300 K, $\tau =\mu \mu_{c}/ev_{f}{}^{2}$ is the Drude relaxation time with the carrier mobility $\mu =1.0\;m^{2}V^{-1}S^{-1}$ and Fermi velocity $v_{f}\approx 10^{6}\;m/s$. In the THz region and below, where the photon energy $\hbar \omega \ll \mu_{c}$, the interband part (Equation (6)) can be neglected compared to the intraband. Therefore, $\sigma_{g}$ can be described by the Drude-like model [56], then Equation (5) can be written as
(7)$$\sigma_{g}(\omega )=\frac{e^{2}\mu_{c}}{\pi \hbar }\frac{i}{\omega +i\tau^{-1}}$$

When τ=0.5 ps,T=300 K, the real and imaginary parts of $\sigma_{g}$ as functions of the chemical potential ($\mu_{c}$) and the wavelength (λ) are displayed in Figure 2a,b. As can be seen from Figure 2, both Re ($\sigma_{g}$) and Im ($\sigma_{g}$) can be dynamically tuned by changing $\mu_{c}$ and λ, because the amplitude modulation and the spectral shift of the resonance are determined by the real and imaginary parts of $\sigma_{g}$, respectively.

For continuous monolayer graphene, the transmission and reflection are expressed as follows [39,57]: (8)$$T={\left|\frac{2}{2+\eta_{0}\sigma_{g}\cos\theta }\right|}^{2}$$
(9)$$R={\left|\frac{\eta_{0}\sigma_{g}\cos\theta }{2+\eta_{0}\sigma_{g}\cos\theta }\right|}^{2}$$
where $\eta_{0}$ is the wave impedance of air; the absorption can be expressed as
(10)A=1−T−R=4η0Re(σg)|2+η0σg|2

In the range of visible light, $\sigma_{g}$ is almost a constant, resulting in only 2.3% absorption. $\sigma_{g}$ varies with the change of τ in the range of terahertz and far infrared. At λ=66.75 μm, τ=0.5 ps, the absorption is 3.9%, while when τ=2.5 ps, the absorption is 0.8%, which is significantly lower than that of patterned graphene. For a symmetric dielectric environment, the predicted maximum absorption would not exceed 50% [43]. The absorption may approach the maximum predicted value when graphene is formed into a specific pattern and/or by adjusting its geometric parameters or the dielectric environment.

## 3. Simulation Results and Discussions

### 3.1. The Influence of Different Chemical Potentials on Absorption

Different from other materials, graphene can be dynamically tunable when its geometry is fixed. This dynamic tenability is achieved by changing $\mu_{c}$. From Equation (7), $\mu_{c}$ can mainly determine $\sigma_{g}$, and $\mu_{c}$ can be tuned and controlled by using an electrostatic field or chemical doping. According to Equation (3), $\mu_{c}$ can reach specific values by applying specific $V_{b}$. By fixing the structure parameters at (*L* = 0.4 μm, *R* = 0.4 μm, *p* = 0.3 μm), the absorption characteristics of different $\mu_{c}$ values are simulated under vertical plane wave illumination. As shown in Figure 3a, when we change $\mu_{c}$ from 0.2 to 0.8 eV, the maximum absorption increases simultaneously, and experiences blue shift. Then, we picked 0.2, 0.4, 0.6, 0.8 eV, and analyzed the absorption spectra. As shown in Figure 3b, the maximum absorption increases and reached 38.2% with absorption spectrum blue shift. Meanwhile, the working bandwidth narrowed gradually with the increase in $\mu_{c}$. Compared with the monolayer unpatterned graphene with an absorption of only 2.3% [58], the maximum absorption improved greatly. We demonstrate the physical mechanism as follows: (1) according to Equation (1), the higher $\mu_{c}$ leads to a higher value of N, the higher value of N contributed to the increase in plasmonic oscillation, and the increase in plasmonic oscillation leads to the enhancement of the maximum absorption; (2) the working bandwidth mainly depends on how fast Z0Re(σg) and Z0Im(σg)+ncotφ change with frequency ω, where $Z_{0}$ and *n* are both constants. For the gradually changing conductivity of continuous graphene film, the fastest changes are related to $\cot\varphi =\cot\left(\frac{\omega }{c}nd_{1}\right)$. It can be seen from Figure 3a that with the increase in $\mu_{c}$, the resonance wavelength (λ) of the corresponding absorption peak gradually decreases. Because λ and ω are inversely proportional, ω gradually increases with the decrease in λ; and since the cotangent function is a monotonously decreasing function, the working bandwidth gradually narrows as ω increases.

Figure 3c shows the local electric field distribution at maximum absorption resonance wavelengths for different $\mu_{c}$ values. As $\mu_{c}$ increases, the intensity of electric field distribution also increases continuously, as well as only being distributed at the edge of the two semi-ellipses; the triangle has no distribution. This is because the two triangles are symmetrical and intersect with each other, thus realizing zero-energy states, and the triangular connection does not play any role in the excitation of plasmons [59]. In addition, the LSPs are enhanced with the increase in $\mu_{c}$; this kind of enhancement leads to a stronger electric field and a higher absorption.

### 3.2. The Influence of Different Semiminor Axes on Absorption

In this paper, the relationships between the elliptical semiminor axis (*R*) and absorption spectra of the structure were also studied. Keeping other parameters unchanged ($\mu_{c}=0.8\;\mathrm{eV},L=0.4\;\mu m,P=3\;\mu m$), *R* was changed from 0.1 μm to 0.4 μm with an interval of 0.1 μm. The results are shown in Figure 4a. With an increase in *R*, the resonance wavelength moves from 42 μm to 29 μm with a significant blue shift. Simultaneously, the maximum absorption increases from 15.2% to 38.2%, a strong enhancement on graphene absorption. 

Figure 4b shows the local electric field distribution at maximum absorption resonance wavelengths for different *R* values. The electric field is distributed on both sides of the semi-elliptical arc, and the intensity of electric field is basically unchanged with the increase in *R*. However, with the increase in *R*, the effective area of graphene increases; the increase in the effective area can make graphene gather more energy, leading to the enhancement in the maximum absorption. The localized surface plasmon resonance (LSPR) on both sides of the semi-elliptical arc is the main contribution to the wavelength position of the spectra; so with the increase in *R*, the resonance wavelength experiences blue shift. Then, we can conclude that with the increase in *R*, the maximum absorption increases and is accompanied by blue shift.

### 3.3. The Influence of Different Semimajor Axes of the Ellipse on Absorption

This paper analyzes the influence of different semimajor axes (*L*) of the ellipse on absorption characteristics. While keeping other parameters unchanged ($\mu_{c}=0.8\;\mathrm{eV},R=0.4\;\mu m,P=3\;\mu m$), *L* of the ellipse was changed. Figure 5a plots the absorption spectra of different *L* values from 0.2 μm to 0.5 μm with an interval of 0.1 μm. As described in the figure, the resonance wavelengths have strong red shifts and the maximum absorption gradually increases. The resonance wavelength shifts from 21.63 μm to 34.07 μm, while the maximum absorption increases from 4.5% to 38.7% and increased nearly 9 times. The increase in the absorption is a result of the graphene surface plasmon resonance (SPR) gradually increasing with the increase in *L*.

The distribution of the electric field intensity at maximum absorptions for different *L* values are shown in Figure 5b. The electric field intensity is only distributed at the edge of the two semi-ellipses. This phenomenon is mainly caused by the accumulation of electric charge. When *L* increases, the distance between adjacent graphene arrays decreases, which leads to the enhancement of coupling between neighboring graphene arrays, and the increase in coupling results in the red shifts and the increase in absorption. Figure 5b also shows that local enhancement of the electromagnetic field can be realized, which is caused by the strong electric dipole resonance excited by the charge at both ends of *L*. This strong resonance can effectively capture the energy of light and have enough time to eliminate the loss of graphene.

### 3.4. The Influence of Different Periods on Absorption

The absorption spectra for different periods (*P*) are plotted in Figure 6a. As can been seen from the figure, *P* has a significant impact on the maximum absorption but less on the wavelength, when other parameters remain unchanged ($\mu_{c}=0.8\;\mathrm{eV},L=0.4\;\mu m,$
R=0.4 μm). As *P* increases, the resonance wavelength first decreases and then is maintained at 28.2 μm; simultaneously, the maximum absorption gradually decreases. The physical mechanisms are explained as follows: (1) with the increases in *P*, the resonance wavelengths are almost unchanged because the resonance condition remains unchanged [39]; (2) the resonance wavelength first decreases at low *P* due to the coupling of neighboring graphene; (3) the filling factor of graphene decreases, leading to the decrease in absorption (the filling factor is defined as the ratio of the graphene resonator to the entire resonant unit cell). 

The distribution of the electric field intensity at maximum absorptions with different *P* values are shown in Figure 6b. The electric field is mainly distributed on both sides and the bottom of the semi-ellipse. As can be seen from Figure 6b, as *P* increases, the electric field intensity concentrated on the ellipse becomes stronger. However, because the filling factor of graphene gradually decreases, and the effect of electric field intensity on the absorption is smaller than that of graphene filling factor on the absorption, the absorption gradually decreases with the increase in *P*.

### 3.5. The Influence of Different Incident Angles on Absorption

We also studied the influence of different incident angles (θ) on the absorption characteristics of the structure with the other parameters unchanged ($\mu_{c}=0.8\;\mathrm{eV},L=0.4\;\mu m,R=0.4\;\mu m,P=3\;\mu m$). Figure 7a,b demonstrate the resonance wavelength under different θ values. As can be seen from the figure, the maximum absorption peaks are sensitive to different θ values and the value decreases with the increase in θ. However, the resonance wavelengths are insensitive to different θ values. The maximum absorption is obtained under the condition of so-called critical coupling or rate equipartition [60]. In the case of oblique incident, the condition of critical coupling cannot be achieved. Furthermore, because *P* is much smaller than the incident wavelength, the resonance frequency is almost independent of the incident angle θ, so the phase difference of the light field at the adjacent graphene structures are basically independent of θ [43]. Figure 7c reveals the corresponding electric field intensity distribution at the resonance wavelength with different incident angles. It can be seen from the figure that as θ increases, the electric field strength decreases continuously and is mainly distributed on both sides of the ellipse.

## 4. Bilayer Graphene Arrays

We also analyzed the influence of bilayer graphene arrays on absorption characteristics. The results show that the maximum absorption can be further improved using the bilayer graphene arrays. As depicted in Figure 8a, the structure consisting of two-layer graphene arrays are separated by a thin SiO_2_ layer with a thickness of $d_{2}$. Whilst keeping other parameters unchanged ($\mu_{c}=0.8\;\mathrm{eV},L=0.4\;\mu m,R=0.4\;\mu m,P=3\;\mu m$), the absorption spectra of various $d_{2}$ values from 100 to 400 nm with 100 nm intervals are illustrates in Figure 8b. When $d_{2}$ is fixed at 100 nm, it can be seen form the spectra that there are dual-band absorption peaks at 21.6 μm and 36.3 μm with an absorption efficiency 41.7% and 11%, respectively. Moreover, when $d_{2}=100\;nm$, the maximum absorption at the short-band can reach 41.7%, which exceeds that of the monolayer graphene arrays with the same parameters (38.2%). The shifts of the resonance wavelength are different for the two bands. For the long-band and short-band, the resonance wavelength undergoes blue shift and slight red shift, respectively. As for the maximum absorption, the maximum absorption at the short-band decreases with the increase in $d_{2}$, but for the long-band, the maximum absorption increases with the increase in $d_{2}$. According to our simulation results, the upper layer of the graphene structure mainly contributes to the short-band, and the lower layer mainly contributes to the long-band. Therefore, the bilayer graphene structure allows us to adjust the two absorption peaks and their bandwidths separately to achieve different absorption characteristics.

## 5. Conclusions

In this paper, we analyze the tunable absorption enhancement of periodic hourglass-shaped GBA. By increasing *R* and *L* of the ellipse, the absorption can be further enhanced. With the increase in *R* and *L*, the absorption experiences red shifts and blue shifts, respectively. The variation of *P* and θ are sensitive to maximum absorption intensity, but insensitive to the resonance wavelength. This paper also presents the absorption characteristics of bilayer graphene, which can bring about higher absorption (41.7%) and even can make dual-band absorption occur. We hope that our work can provide a potential application in graphene-based optoelectronic devices, biosensors, and environmental monitoring.

## Figures and Tables

**Figure 1 nanomaterials-10-00533-f001:**
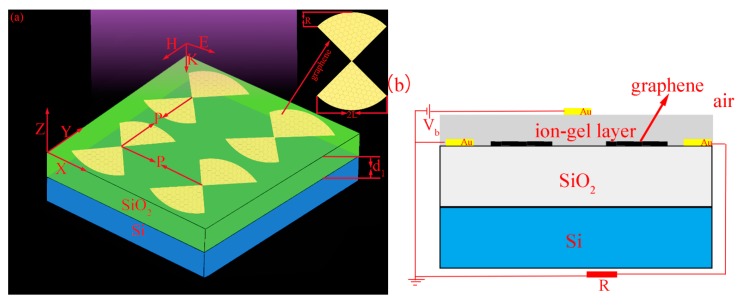
(**a**) The schematic of “hourglass” graphene arrays structure with period (*P*), semimajor axis (*L*), semiminor axis (*R*). The two layers of substrate structure are Si and SiO_2_. The thickness of SiO_2_ is $d_{1}$. (**b**) The side view of the structure which manipulates the chemical potential ($\mu_{c}$) of graphene by applying a voltage ($V_{b}$).

**Figure 2 nanomaterials-10-00533-f002:**
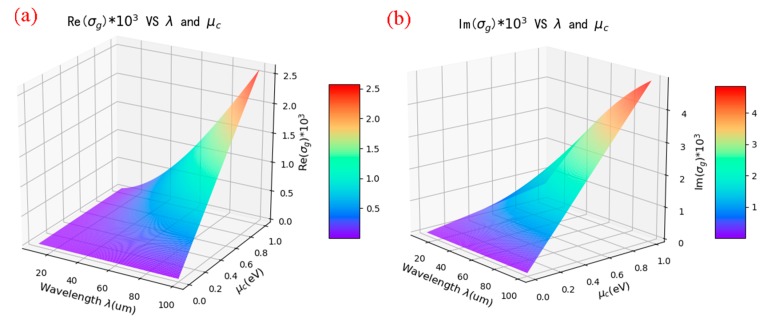
The real (**a**) and imaginary part (**b**) of $\sigma_{g}$ as functions of $\mu_{c}$ and λ.

**Figure 3 nanomaterials-10-00533-f003:**
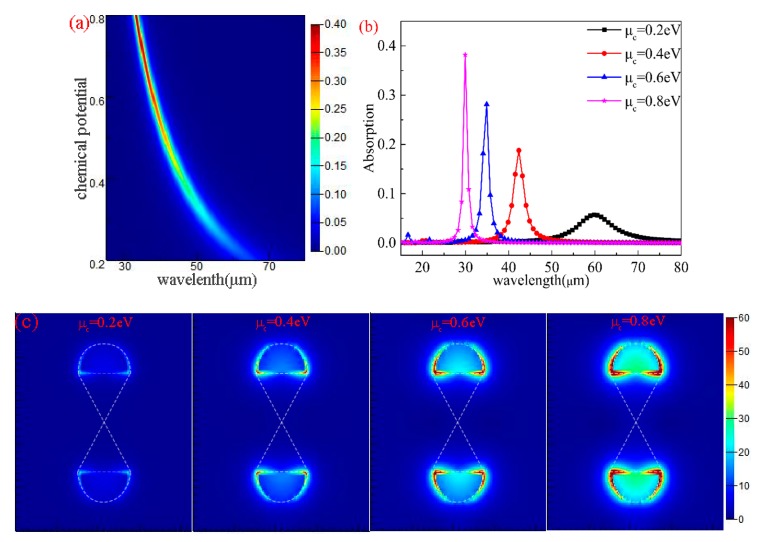
(**a**) The graph of the graphene absorption. (**b**) The absorption spectra corresponding to the chemical potential from 0.2 to 0.8 eV. (**c**) The corresponding electric field intensity distribution under different $\mu_{c}$ values.

**Figure 4 nanomaterials-10-00533-f004:**
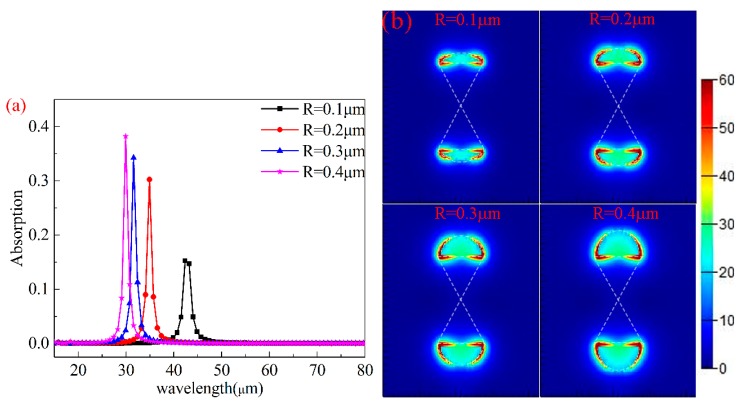
(**a**) The graph of the graphene absorption corresponding to the semiminor axis from 0.1 to 0.4 μm. (**b**) The corresponding electric field intensity distribution under different *R* values.

**Figure 5 nanomaterials-10-00533-f005:**
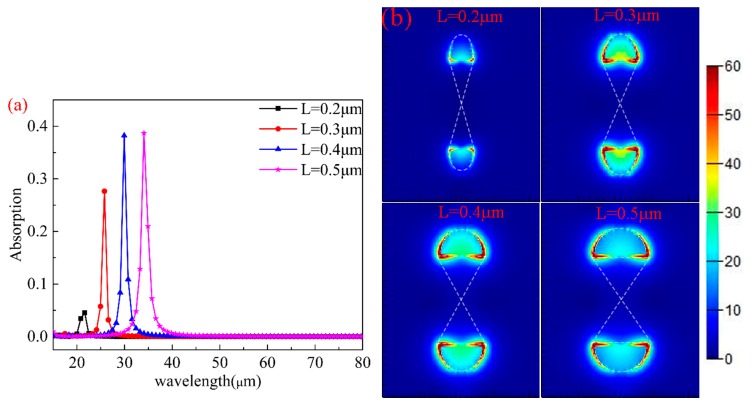
(**a**) The absorption spectra of the structure with different *L* values. (**b**) The electric field distributions at the maximum absorption for L=0.2,0.3,0.4,0.5 μm.

**Figure 6 nanomaterials-10-00533-f006:**
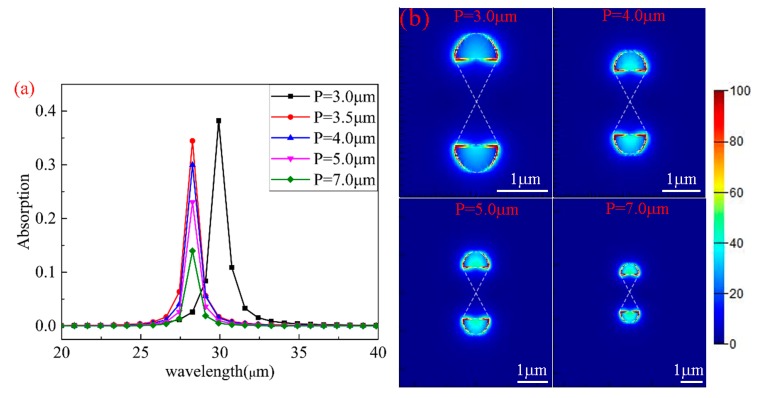
(**a**) The absorption spectra of the structure with different *P* values. (**b**) The electric field distributions at the absorption peak for P=3.0,4.0,5.0,7.0 μm.

**Figure 7 nanomaterials-10-00533-f007:**
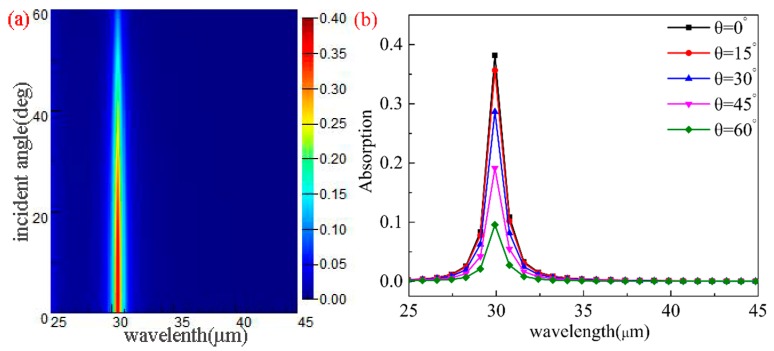
(**a**,**b**) The absorption spectra of the structure with different θ values. (**c**) The electric field distributions at the absorption peak for $\theta =0^{\circ },30^{\circ },45^{\circ },60^{\circ }$.

**Figure 8 nanomaterials-10-00533-f008:**
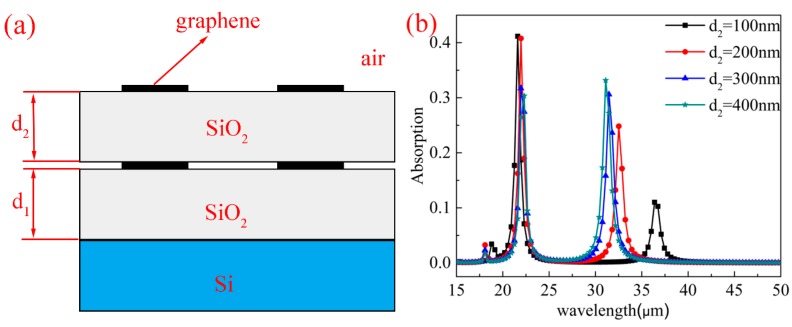
(**a**) The side view of the structure consisting of bilayer graphene arrays covered with two thin SiO_2_ layers with thickness $d_{1}$ and $d_{2}$. (**b**) The absorption spectra of the structure with different $d_{2}$ values.

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
