# Peer review of "A Tunable Terahertz Metamaterial Absorber Composed of Hourglass-Shaped Graphene Arrays"

_nanomaterials, 2020, doi:10.3390/nano10030533_

Round 1

Reviewer 1 Report

The paper is interesting. They propose periodically patterned hourglass graphene to increase the absorption and show an exhaustive numerical results. 

Author Response

Response 1:  Thank you for your recognition of our work, we are honored that our work interest you, and we will continue to move forward in this area. Thank you again for your review of our paper.

For our work, your opinion has helped us a lot. Thank you again for your review of our paper.

Reviewer 2 Report

In this manuscript, the authors have studied the possibility of tunable absorption enhancement in the periodic hourglass-shaped
graphene-based absorbers. This was done by screening the influence of the geometrical and chemical properties of the graphene-mediated structure on the absorption spectra. By using numerical investigations, the spectral characteristics of the proposed platforms was studied in details. Although, the work comprehensively considered the properties and performance of the graphene-assisted absorber for THz and infrared bands, it suffers from important shortcomings that must be answered. I listed the issues below and do suggest the authors to carefully address these comments in the revisions.

General comments:

1) The quality of writing is too poor and must be improved carefully. There are obvious grammatical mistakes along the manuscript. In addition, there are some sentences out of scientific fashion and must be revised or deleted.

2) Given that FDTD model was conducted to define the properties of the structure, the utilized commercial package must be cited.

3) The use of graphene monolayer in the development of THz metadevices must be included in the introduction part of the manuscript, for instance, detectors (Nat. Mater. 2012, 11(10), 865-871), modulators (Nanoscale 2019, 11(17), 8091-8095 and ACS Photonics 2015, 2(11), 1559-1566), etc.

Moreover, there are some important and good review articles that must be included in the references, such as: Carbon 2015, 82, 229-237, ACS Nano 2014, 8(2) 1086-1101, and IEEE J. Selected Topics in Quantum Electronics 2013, 20(1), 130-138, etc.

Specific comments:

1) The performance of the proposed graphene-mediated absorber structure must be compared with the ones proposed based on all-dielectric metastructure and also plasmonic ones (Adv. Opt. Mater. 2015 3(3), 376-380). This allows the readers to understand how efficient is the graphene-assisted device. Also, the advantages of the devised platform should be compared with such designs.

2) The computed E-field maps are not consistent with the shape of the design. This must be explained. In general, it seems that the E-field snapshots were taken for another structure. Also, there is no any single field confinement at the center of the design, why?

3) The realization of the multilayer structure in Figure 8, seems to be extremely challenging. The fabrication of such a design requires multistep, costly, and time consuming techniques. Indeed, the actual purpose of the design is not clear at all. If the goal is broad band or multiresonant absorption feature, there are promising alternatives in the literature. 

Round 2

Reviewer 2 Report

Publishable as is.

Reviewer 3 Report

The Authors modified the manuscript accordinlgy and gave satisfactory replies to my comment. Therefore, I suggest to accept the amended manuscript.